# Sonic Hedgehog Signature in Pediatric Primary Bone Tumors: Effects of the GLI Antagonist GANT61 on Ewing’s Sarcoma Tumor Growth

**DOI:** 10.3390/cancers12113438

**Published:** 2020-11-19

**Authors:** Mathilde Mullard, Marie Cadé, Sarah Morice, Maryne Dupuy, Geoffroy Danieau, Jérome Amiaud, Sarah Renault, Frédéric Lézot, Régis Brion, Rose Anne Thepault, Benjamin Ory, François Lamoureux, Isabelle Corre, Bénédicte Brounais-LeRoyer, Françoise Rédini, Franck Verrecchia

**Affiliations:** 1INSERM, UMR1238 “Bone Sarcomas and Remodeling of Calcified Tissues”, Nantes University, F44035 Nantes, France; mathilde.mullard@univ-nantes.fr (M.M.); sarah.morice@univ-nantes.fr (S.M.); maryne.dupuy@etu.univ-nantes.fr (M.D.); geoffroydn@gmail.com (G.D.); jerome.amiaud@univ-nantes.fr (J.A.); sarah.renault@univ-nantes.fr (S.R.); frederic.lezot@univ-nantes.fr (F.L.); regis.brion@univ-nantes.fr (R.B.); Rose-Anne.Thepault@univ-nantes.fr (R.A.T.); Benjamin.Ory@univ-nantes.fr (B.O.); francois.lamoureux@univ-nantes.fr (F.L.); isabelle.corre@univ-nantes.fr (I.C.); benedicte.brounais@univ-nantes.fr (B.B.-L.); francoise.redini@univ-nantes.fr (F.R.); 2Centre de Recherche en Cancérologie et Immunologie Nantes Angers, INSERM UMR1232, Nantes University, F44035 Nantes, France; marie.cade@etu.univ-nantes.fr; 3Institut de Cancérologie de l’Ouest, Saint Herblain, F-44805 Nantes, France; 4CHU Hôtel Dieu, F44035 Nantes, France

**Keywords:** SHH, Gli1, GANT61, ewing’s sarcoma, osteosarcoma, primary tumor growth

## Abstract

**Simple Summary:**

The poor clinical outcomes for Osteosarcoma (OS) and Ewing’s sarcoma (ES) patients underscore the urgency of developing novel therapeutic strategies for these pathologies. In this context, the emerging role of Sonic hedgehog (SHH) signaling in cancer has been critically evaluated, focusing on the potential for targeting SHH signaling as an anticancer strategy. The aims of this work were (1) to highlight and to compare a possible SHH/Gli signature between OS and ES, (2) to strengthen our knowledge concerning the role of EWS-FLI1 in the SHH signature in ES and (3) to evaluate the effect of the specific Gli inhibitor GANT61 in vivo on the growth of ES tumors using an orthotopic mice model. Our work identifies Gli1 as a promising therapeutic target in ES and demonstrates that GANT61, through inhibition of Gli1 transcriptional activity, may be a promising therapeutic strategy hindering ES tumor progression, and specifically primary tumor growth.

**Abstract:**

Osteosarcoma (OS) and Ewing’s sarcoma (ES) are the most common malignant bone tumors in children and adolescents. In many cases, the prognosis remains very poor. The Sonic hedgehog (SHH) signaling pathway, strongly involved in the development of many cancers, regulate transcription via the transcriptional factors Gli1-3. In this context, RNAseq analysis of OS and ES cell lines reveals an increase of some major compounds of the SHH signaling cascade in ES cells, such as the transcriptional factor Gli1. This increase leads to an augmentation of the transcriptional response of Gli1 in ES cell lines, demonstrating a dysregulation of Gli1 signaling in ES cells and thus the rationale for targeting Gli1 in ES. The use of a preclinical model of ES demonstrates that GANT61, an inhibitor of the transcriptional factor Gli1, reduces ES primary tumor growth. In vitro experiments show that GANT61 decreases the viability of ES cell, mainly through its ability to induce caspase-3/7-dependent cell apoptosis. Taken together, these results demonstrates that GANT61 may be a promising therapeutic strategy for inhibiting the progression of primary ES tumors.

## 1. Introduction

Osteosarcoma (OS) and Ewing’s sarcoma (ES) are the most frequently observed primary malignant bone tumors in the second decade of life [1]. Conventional treatment consists of complete surgical resection combined with neo-adjuvant and adjuvant chemotherapies [2,3,4,5]. Resistance to treatment remains, thus, one of the main causes of death in these patients with approximately a 5-year survival rate of 70–75% for localized disease and patients who are good responders to chemotherapy compared to 20% for patients resistant to chemotherapy.

Regarding OS, whole-genome sequencing analyses demonstrate that OS are characterized by very high rates of genetic alterations with somatic mutations and copy number alterations [6]. Concerning ES, a translocation t(11;22)(q24;q12) is observed in 85% of cases and leads to the production of the fusion protein EWS-FLI1 [7,8]. The lack of response to conventional treatments reported in many OS and ES patients underscores the urgency of developing novel therapeutic strategies. In this context, recent progress in understanding the molecular basis of the pathogenesis of OS and ES, and the emergence of therapies developed to specifically inhibit signaling pathways associated with cancer are promising [9,10].

The Sonic hedgehog (SHH) signaling pathway is a conserved pathway that regulates developmental processes [11]. The cellular response to SHH can be achieved through two pathways, the canonical and the non-canonical. The first one is based on the binding of SHH to the receptor Patch (Ptch) [12], which continuously inhibits the activity of the G-coupled receptor Smoothened (SMO) in the absence of ligand. Following the binding of its ligand, Ptch is degraded [13], resulting in the activation of SMO. This activation induces the dissociation of the complex formed by Gli and suppressor of fused homolog proteins (Sufu). This dissociation stimulates the translocation of Gli proteins (Gli1, 2 or 3) into the nucleus, where they act as transcriptional factors [14,15,16]. The main mechanism of non-canonical signaling of SHH involves the activation of Gli independently of SHH itself or of Ptch or SMO. For example, Gli1 and Gli2 are transcriptional targets of Transforming Growth-Factor-β/Smad3 in melanoma [17,18,19]. During the last years, it has been shown that the SHH/Gli signaling pathway is involved in many human cancers [20,21,22,23,24]. The mutation of some genes of the SHH cascade, such as SMO, Ptch and Sufu, have, thus, been associated with the development of various tumors [25,26]. However, the SHH pathway can also be activated without these gene mutations, mainly through a paracrine effect of SHH. Schematically, the Gli transcriptional factors may stimulate the ability of tumor cells to proliferate, thus increasing tumor growth [20,21,22,23,24].

Literature is scarce concerning the SHH pathway and primary bone sarcomas, specifically ES. Gli1 seems particularly important, as it has been identified as a direct transcriptional target of EWS-FLI1 by a mechanism independent of the SHH ligand [27]. Although more than 50 inhibitors of the SHH cascade have been synthetized during recent decades [11,28], such as the SMO receptor inhibitor cyclopamine or Gli inhibitors such as GANT58 and GANT61, none of these specific inhibitors have so far been evaluated on a preclinical model of ES. A unique study using arsenic trioxide, a non-specific inhibitor of Gli1/2, in a xenograft mice model demonstrated the rationale of targeting the SHH cascade in order to block ES tumor growth [29].

In this context, the aims of this work were (1) to highlight and to compare a possible SHH/Gli signature between OS and ES, (2) to strengthen our knowledge concerning the role of EWS-FLI1 in the SHH signature in ES and (3) to evaluate the effect of the specific Gli inhibitor GANT61 in vivo on the growth of ES tumors using an orthotopic mice model.

## 2. Results

### 2.1. Elevation of Gli1 Target Gene Expression in ES Cell Lines Compared to Osteosarcoma (OS) Cell Lines

To better understand the role of the SHH pathway in the development of primary bone tumors, we first compared the gene expression of the main components of the SHH cascade in seven ES cell lines with those measured in seven OS cell lines, using RNAseq analysis. As shown in Figure 1A, the RNAseq analysis showed that multiple actors of the SHH cascade are significantly overexpressed in ES cells compared to OS cells. These genes include the transcriptional factor Gli1 and some receptors of the SHH cascade, such as Ptch1 and SMO. To confirm these results, we performed a RT-qPCR analysis. As shown in Figure 1B,C, the expression of several compounds of the SHH cascade, such as Gli1 (Figure 1B) and Ptch1, Ptch2 and SMO (Figure 1C), were significantly increased in ES cell lines compared to OS cell lines. Note that the EW24 ES cells appeared to have lower Gli1 expression than the other ES cell lines (Figure 1A). However, the data obtained using RT-qPCR indicated that the Gli1 expression in this cell line remained higher than those measured in all the OS lines except the U2OS cells. In contrast to Gli1, the expression of Gli2 and Gli3 was not modified (Figure 1B). To determine whether the acquisition of this SHH/Gli signature in ES cells leads to an increase in the transcriptional response of Gli, we then performed promoter gene reporter assays using the specific Gli-reporter construct Gli-lux. As shown in Figure 2A, transactivation of the Gli-lux construct was greatly increased in each ES cell line compared to each OS cell line. In addition, RNAseq analysis showed high expression of various Gli1 target genes in ES cell lines compared to OS cell lines (Figure 2B). These genes include Stathmin 1 (Stmn1) and NK2 homeobox 2 (NKx2.2), previously described as SHH target genes in tumor tissues [30]. The RT-qPCR analysis clearly demonstrated that even for EW24 ES cells that express Gli1 more weakly, the level of Stmn1 and NKx2.2 expression remains higher than that measured in OS cells (Figure 2C).

Together, these results demonstrated that the Gli1 signature identified in ES cells leads to an increased Gli transcriptional response in ES cell lines.

### 2.2. EWS-FLI1 Drives the Expression of Gli1 and the Gli Transcriptional Response in ES

Since Beauchamp and colleagues [27] described Gli1 as a direct target of the fusion protein EWS-FLI1, we measured the effects of EWS-FLI1 silencing on the transcriptional response of Gli1 using ES A673 cells stably transfected with a doxycycline inducible shRNA directed against EWS-FLI1. As shown in Figure 3, the treatment of these cells with doxycycline induces a decrease in the mRNA level of EWS-FLI1 (Figure 3A). As expected, the decrease in EWS-FLI1 mRNA levels led to a decrease in Gli1 expression (Figure 3B). To determine whether this effect of EWS-FLI1 on Gli1 expression leads to modulation of the transcriptional response of Gli1, cells were transiently transfected with the Gli-specific promoter gene/reporter Gli-lux. As shown in Figure 3C, the treatment of cells with doxycycline induced a decrease in the transactivation of the Gli-lux construct. In addition, RT-qPCR analysis demonstrates that the expression of various Gli target genes such as Ptch1, Ptch2, Stmn1 and Nkx2.2 was significantly reduced when EWS-FLI1 expression was reduced (Figure 3D–G).

Together, these results, in accordance with the previous ones published by Beauchamp and colleagues [27], demonstrated that EWS-FLI1 plays a crucial role in the transcriptional response driven by Gli1 in ES cells.

### 2.3. GANT61 Inhibits the Gli Signaling Pathway and Primary Tumor Growth in an Orthotopic Model of ES

As we demonstrated that the intrinsic activation of SHH signaling was mainly due to Gli1 overexpression in response to EWS-FLI1, we then assessed the effects of the specific Gli inhibitor GANT61 on primary ES tumor growth. The efficiency of GANT61 was first evaluated by Gli-lux gene reporter assay in TC71, A673 and SKES1 ES cell lines. GANT61 decreased Gli-dependent luciferase activity in the three cell lines (Figure 4A). In addition, treatment of the cells with GANT61 significantly decreased the mRNA expression of two Gli1 target genes, Stmn1 and Kx2.2 (Figure 4B).

The crucial role played by the bone microenvironment in the growth of ES tumors led us to choose to use an orthotopic mouse model to evaluate the effect of GANT61 on the growth of ES tumors. As shown in Figure 4C, treatment of mice with GANT61 significantly reduced tumor growth. Indeed, the mean tumor size at day 21 was 2290 ± 627 mm^3^ when the mice were treated with vehicle (control group) and only 1198 ± 298 mm^3^ when the mice were treated with 50 mg/kg of GANT61.

Together, these results demonstrated the ability of GANT61 (1) to block intrinsic activation of Gli signaling in ES cell lines and (2) to inhibit the primary tumor growth of ES in an orthotopic model of ES.

### 2.4. GANT61 Induces In Vitro Cell Death

To better understand the mechanisms that control the effects of GANT61 on ES tumor growth, we then performed in vitro experiments. Firstly, our experiments show that GANT61 significantly inhibited the viability of three ES cell lines in a dose-dependent manner (Figure 5A). Secondly, we studied whether the reduction in ES cell survival by GANT61 is associated with the induction of cell apoptosis. Using flow cytometric annexin V/PI assays, we demonstrated that GANT61 induces early and late apoptotic events in a dose-dependent manner (Figure 5B). For example, the percentage of A673 cells in early apoptosis (Annexin V+/PI−) was 4.3% ± 1.2% in the absence of GANT61, and reached 10.3% ± 2.2% and 14.9% ± 12.2% after 24 h treatment of cells with 5 µM and 15 µM GANT61, respectively. The percentage of A673 cells in late apoptosis (Annexin V+/PI−) was 16.5% ± 4.6% in non-treated condition but reached 21.7% ± 8.1% and 36.7% ± 4.2% after 24 h of treatment of the cells with 5 µM and 15 µM GANT61, respectively. Noticeably and unlike SKES1 and A673, TC71 cells appear to be less sensitive to GANT61. Thirdly, we measured the activity of apoptotic enzymes caspases-3 and -7 in ES cells treated 24 h with GANT61. As shown in Figure 5C, GANT61 significantly stimulated the activity of caspase-3 and -7. Thirdly, our experiments demonstrated the dose-dependent cleavage of PARP (Poly(ADP-ribose) polymerase) in all three ES cell lines (Figure 5D).

Taken together, these results demonstrate that the Gli inhibitor GANT61 reduces the viability of ES cell lines in vitro and suggest that this effect is mainly due to its ability to induce caspase-3/7-dependent apoptosis.

## 3. Discussion

The poor clinical outcomes for OS and ES patients underscores the urgency of developing novel therapeutic strategies for these pathologies. In this context, the role of the SHH signaling pathway in cancer has been evaluated, focusing on the potential for targeting the actors of this signaling pathway as an anti-cancer strategy [31]. For OS, the consequences of an alteration of the SHH pathway in tumor development have been evaluated in several cell models and in preclinical animal models [32]. For example, Mohseny et al. showed activation of the SHH pathway in various OS cell lines without correlating these results with patient survival [33]. Thus, analysis of the expression of various components of the SHH pathway in five OS cell lines (NHost, 143B, HOS, MG63 and NOS-1) shows a strong expression of SHH, Ptch1, SMO, Gli1 and Gli2 [34]. However, only an increase in SHH, Ptch 1 and Gli2 was found in patient biopsies [34]. Regarding the transcriptional factors Gli1-3, most studies have focused on the role of the transcription factor Gli2 in OS tumor development and progression [11]. Most work suggests that activation of the SHH signaling pathway in OS is closely related to the binding of the SHH ligand on membrane receptors [32]. Thus, many studies have targeted the SMO receptor by chemical or molecular approaches in OS. For example, Hirotsu et al. showed that cyclopamine, an SMO inhibitor, and a shRNA against SMO both inhibit the proliferation of 143B and HOS OS cells and the growth of the primary tumor [34]. One recent study demonstrated that the Gli1/2 inhibitor GANT61 is able to inhibit the growth of OS tumors by inducing oxidative stress via the miRNA-1286/RAB31 axis using an OS xenograft model [35].

By comparing the expression of the main components of the SHH pathway among seven OS cell lines and seven ES cell lines, our work shows that the expression of Gli1, Ptch1, Ptch2 and SMO is strongly increased in the ES cell lines. Ptch1 and Ptch2 have been defined as targets of Gli [36,37]. This increase in Gli1 expression in ES cell lines results in an increased transcriptional response, as demonstrated by the vector/reporter gene experiments and the evaluation of the Gli1 target gene response. If several studies have shown the interest of targeting the SHH pathway in OS [32], our work shows that this interest is even greater in ES, since Gli1 expression and its transcriptional activity are dramatically enhanced.

The increased Gli expression and its transcriptional activity may be independent of the binding of the SHH ligand to its receptors Ptch and SMO [11]. For example, the epidermal growth factor receptor (EGFR) activates Gli via its ability to stimulate the extracellular signal-regulated kinase pathway (ERK) during tumor development [38]. Ras signaling induces Gli1 gene expression in gastric cancer [39]. The TGF-β/Smad3 signaling pathway is also able to enhance Gli2 expression in melanoma [17,18,19]. Consistent with the previous results, which demonstrate that Gli1 is a direct transcriptional target of EWS-FLI1 in ES [27], our work now shows that EWS-FLI1 regulates Gli1 expression and its transcriptional activity independently of the SHH canonical pathway. Our work reinforces the crucial role of EWS-Fli1 in the development of ES as previously demonstrated. Indeed, the reduction of EWS-FLi1 expression by small interfering RNA significantly reduces the proliferative, invasive and tumorigenic phenotype of ES, both in vitro and in vivo [40,41]. Thus, since the transcriptional activity of Gli1 is directly activated by EWS-Fli1 in ES, it appears crucial to directly target the transcription factor Gli1, rather than targeting the SMO receptor, using cyclopamine, for example.

To our knowledge, no studies have yet evaluated the anti-tumor activities of a direct Gli1 inhibitor such as GANT61 in ES. One study using arsenic shows the rationale for targeting the SHH pathway in ES [29]. However, arsenic targets not only the SHH/Gli pathway but also various other cellular pathways, such as those dependent on JNK (Jun-N-terminal Kinase), NFκB (Nuclear Nactor-kappa B) or various MAPKs (Mitogen Activates Protein Kinases) [42].

In the present work, we show for the first time the in vivo anti-tumor effects of a specific Gli1-2 inhibitor, GANT61 [43,44], on ES tumor growth using an orthotopic mouse model. The choice of this model appears crucial since it reproduces the development of ES tumor in humans by integrating signals emanating from the bone microenvironment, contrary to other models such as subcutaneous or intramuscular mice models. In addition, in vitro experiments demonstrated that GANT61 decreases TC71, SKES1 and A673 cell viability at concentrations of 5 μM to 15 μM for 48 h treatments. These values are consistent with many previous studies showing that the cytotoxic effects of GANT61 are observed in the vast majority of cancer cells with IC50 values ranging from 5 μM to 15 μM [31]. Numerous studies have shown that the cytotoxic effect of GANT61 is closely related to cell death rather than being a direct effect on cell proliferation or the cell cycle [31]. The present work demonstrates that GANT61 decreases ES cell viability, mainly through a caspase-3/7/PARP-dependent cell apoptosis pathway, and in accordance with previous studies conducted in neuroblastoma, squamous lung cancer or gastric cancer cells [45,46,47]. Interestingly GANT61 seems also to be able to overcome the resistance of rhabdomyosarcoma and ES cells to chemotherapy such as vincristine [48].

## 4. Materials and Methods

### 4.1. Cell Cultures

Human OS (HOS, KHOS, G292, MG63, U2OS, SJSA1 and SAOS2) and ES (TC71, A673, MHHES1, EW24, RDES, SKES1 and EW3) cell lines were cultured in Dulbecco’s Modified Eagle’s Medium (Lonza, Basel, Switzerland) or Roswell Park Memorial Institute medium (Lonza) in presence of 10% fetal bovine serum (Hyclone Perbio, Bezons, France) and 1% penicillin-streptomycin (Lonza, Basel, Switzerland). The A673 ES cell line was stably transfected to integrate an inducible shRNA against EWS–FLI1 onto its genome [49]. Treatment with doxycycline (1 μg/mL) was sufficient to induce the shRNA expression (A673-1c cells). Authentication of cell lines used in the research has been realized by PCR-single-locus-technology (Eurofins, Ebersberg, Germany). Mycoplasma level has been tested according manufacturer protocol (Lonza, Basel, Switzerland). GANT-61, purchased from Tocris Bioscience (Bristol, UK), was dissolved in dimethyl sulfoxide and ethanol for the in vitro and in vivo studies, respectively.

### 4.2. Real-Time Polymerase Chain Reaction

Total RNA from each OS and ES cell lines was extracted using NucleoSpin^®^RNAII (Macherey Nagel, Duren, Germany). Total RNA was used for first-strand cDNA synthesis using the ThermoScript RT-PCR System (Invitrogen, Carlsbad, CA, USA). Real-time PCR was performed with a CFX 96 real-time PCR instrument (Biorad, Richmond, CA, USA) using SYBR Green Supermix reagents (Biorad). Primer sequences are provided in Table 1.

### 4.3. RNA Sequencing and Analysis

Library preparation and sequencing were performed at Active Motif, Inc. Libraries were prepared from purified RNA using the illumina TruSeq Stranded mRNA Sample Preparation kit, and sequencing was done on the illumina NextSeq 500 as 42-nt long-paired end reads. Read mapping and fragment quantification for each gene were also performed at Active Motif. Read mapping against the human genome (GRCh38) was done using the STAR algorithm with default settings, and fragment assignment was done using feature counts with gene annotations from the Sub-read package. Only read pairs having both ends aligned with a minimum overlap of 25 base pairs, and mapping to the same chromosome and on the same strand, were counted (feature counts -p -B -C—min Overlap 25). Differential gene expression analysis was performed using the DESeq2 package. The *p*-values obtained were corrected for false positives by using the independent hypothesis weighting (package IHW) multiple testing adjustment method. Genes were considered significantly differentially expressed if the log2 fold change was over 1 or less than −1 and if FDR was less than 0.05.

### 4.4. Transient Cell Transfections, Reporter Assays and Plasmid Constructs

OS and ES cell were transfected with jetPEI^TM^ (Polyplus-transfection, Illkirch, France) according to the manufacturer’s recommendations. Transfection efficiencies was monitored using phRLMLP-Renilla luciferase expression vector. Luciferase activity was determined using the LightSwitchTM Luciferase Assay system (Active Motif, Carlsbad, CA, USA). The GLI-Lux construct was used as a reporter construct specifically for Gli-driven signaling (LightSwitch™ Synthetic Response Elements, Active Motif).

### 4.5. Proliferation Assay

ES cell lines were seeded in a 96-well plate. After adherence, cells were treated with increasing concentrations of GANT61 for 48 h. Cells were then fixed with 1% glutaraldehyde and stained using crystal violet. Finally, the crystal violet staining was solubilized in Sorenson solution. Absorbance was measured at 570 nm with a Victor^2^ apparatus (Perkin Elmer, Villebon-sur-Yvette, France).

### 4.6. Annexin V Assay and Caspase Activity

ES cells were cultured and treated with or without GANT61 for 24 h. Cell apoptosis were evaluated by flow cytometry (Cytomics FC500; Beckman Coulter, Roissy, France) according the FITC Annexin V Apoptosis Detection Kit I protocol (BD Biosciences). Caspase3/7 activity was measured in protein total extract using the Apo-ONE caspase 3/7 kit from Promega (Charbonnière les bains, France).

### 4.7. Western Blot Analysis

Cells were lysed in lysis buffer (Sodium Sodecyl Sulfate (SDS) 1%, Tris pH 7.4 10 mM, Sodium orthovanadate 1 mM). Protein concentration was determined using the BCA Protein Assay Kit (Sigma-Aldrich, St Quentin-Fallavier, France). Samples in Laemmli buffer (62.5 mM Tris–HCl, pH 6.8, 2% SDS, 10% glycerol, 5% 2-mercaptoethanol and 0.001% bromophenol blue) containing equal amounts of total protein extracts were separated by SDS-polyacrylamide gel electrophoresis (SDS-PAGE). After transfer to PVDF (PolyVinyliDene Fluoride) membranes (Thermo Scientific, Illkirch, France), these were probed 12 h with primary antibodies PARP and β-tubulin (Cell signaling, Leiden, The Netherlands). Membranes were finally probed with secondary fluorescent antibodies (Li-COR IRDye). Antibody binding was visualized with the LI-COR odyssey Fc system (Cambridge, UK).

### 4.8. Ewing’s Sarcoma (ES) Mouse Model

Five-week-old female Rj:NMRI nude mice (Elevages Janvier, Le Genest Saint Isle, France) were used for in vivo experiments in accordance with the institutional guidelines of the Ethical Committee (CEEA Pays de la Loire no.06) and the authorization of the French Ministry of Agriculture and Fishery (Apafis # 8405-2017010409498904). Anesthetized mice received a paratibial injection of 1 × 10^6^ TC71 cells, leading to a rapidly growing tumor in bone tissue. Once their tumors were palpable, mice were randomly assigned to control (vehicle) or GANT61 groups. Tumor volume (V) was measured twice a week. Notably, no major changes were observed following injection of the drug GANT61 (weight loss or change in mice behavior) suggesting good tolerance to the drug.

### 4.9. Statistical Analysis

Statistical analyses were performed using GraphPad Prism 6 software (GraphPad Software, La Jolla, CA, USA). The Mann–Whitney test was used for in vitro and in vivo analyses. Results are given as means ± SD for in vitro experiments and as means ± SEM for in vivo experiments. Results with *p* < 0.05 were considered significant.

## 5. Conclusions

Altogether, these results identify Gli1 as a promising therapeutic target in Ewing sarcoma and demonstrate that GANT61, through inhibition of Gli1 transcriptional activity, may be a promising therapeutic strategy hindering ES tumor progression, specifically primary tumor growth.

## Figures and Tables

**Figure 1 cancers-12-03438-f001:**
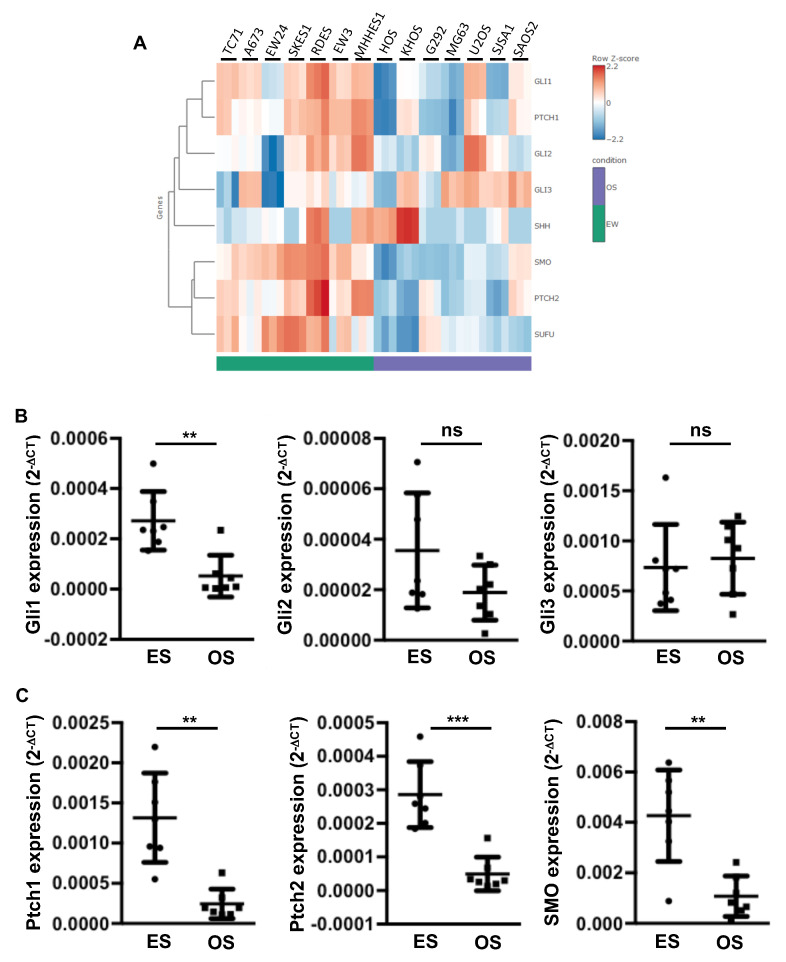
Increased expression of the Sonic hedgehog (SHH) cascade compounds in Ewing’s sarcoma (ES) cell lines. (**A**) heatmap showing color-coded expression of SHH cascade compounds in seven osteosarcoma (OS) cells and seven ES cells following bioinformatics analysis of RNAseq data. High expression (red); low expression (blue). (**B**,**C**) Gli1, Gli2, Gli3 (**B**) and Patch (Ptch)1, Ptch2 and Smoothened (SMO) (**C**) mRNA steady-state levels were quantified by RT-qPCR analysis of seven OS cells and seven ES cells (each point represents the value of one cell line, bars indicate means ± SD of 3 independent experiments, each performed in triplicate, ** *p* < 0.005; *** *p* < 0.001).

**Figure 2 cancers-12-03438-f002:**
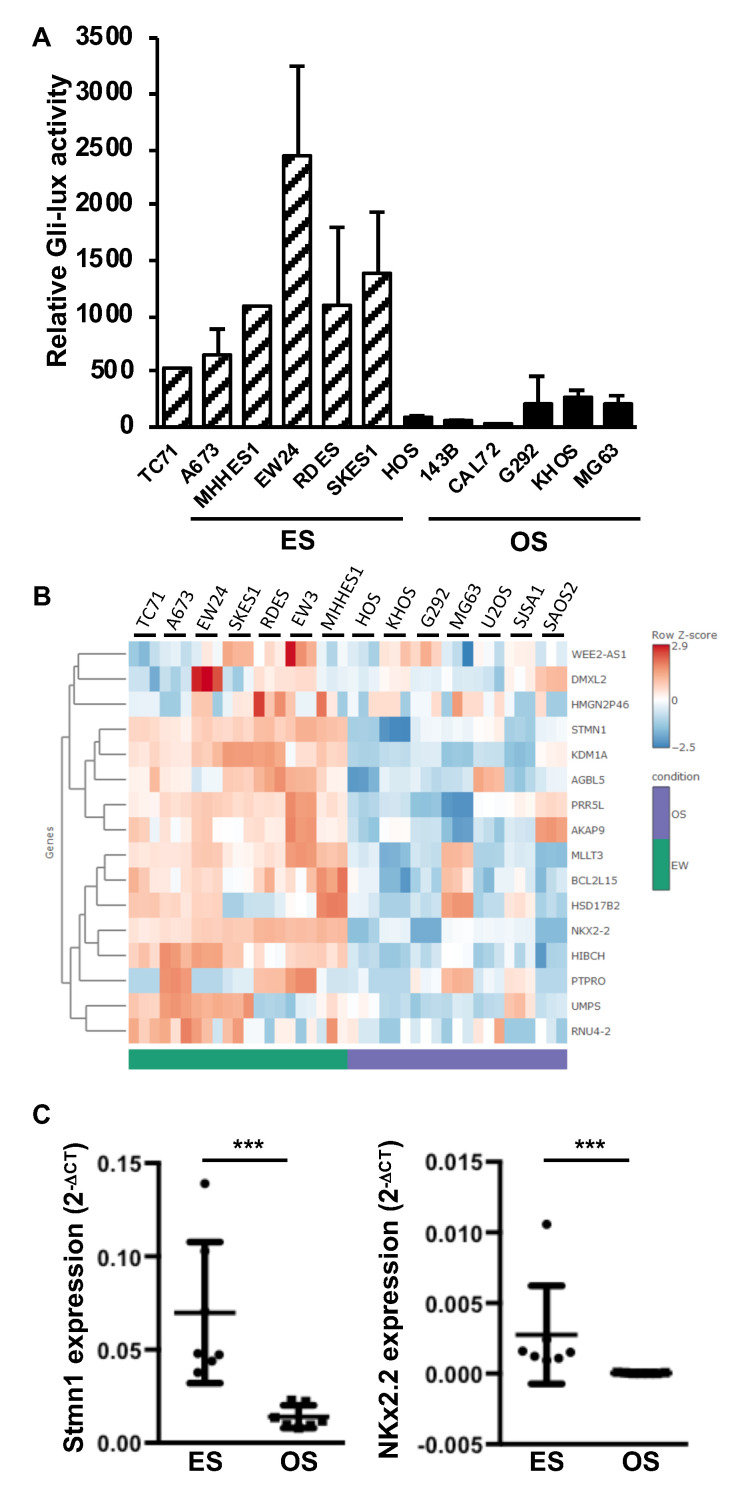
Elevation of Gli1 target gene expression in ES cell lines. (**A**) six ES cells (TC71, A673, MHHES1, EW24, RDES and SKES1) and six OS cells (HOS, 143B, CAL72, G292, KHOS and MG63) were transiently transfected with the Gli-lux construct. Bars indicate means ± SD of 3 independent experiments, each performed in duplicate. (**B**) heatmap showing color-coded expression of SHH target genes in six OS cells and six ES cells following bioinformatics analysis of RNAseq data. High expression (red); low expression (blue). (**C**) Stmn1 and NKX2.2 mRNA steady-state levels were quantified by RT-qPCR analysis of seven OS cells and seven ES cells (each point represents the value of one cell line, bars indicate means ± SD of three independent experiments, each performed in triplicate, *** *p* < 0.001).

**Figure 3 cancers-12-03438-f003:**
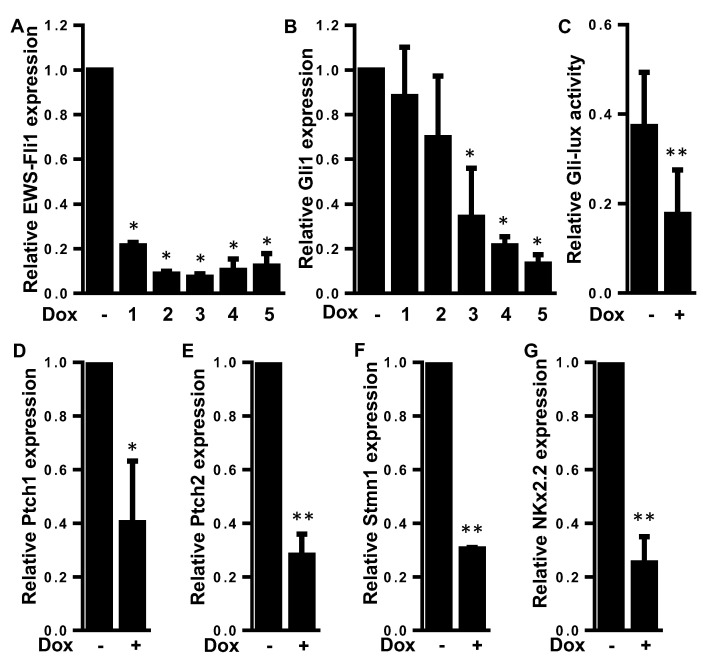
EWS-FLI1 drives the expression of Gli1 and the Gli transcriptional response in ES. (**A**,**B**) A673-1c ES cells were treated or not with doxycycline (1 μg/mL) during 1 to 5 days. EWS-FLI1 (**A**) and Gli1 (**B**) mRNA steady-state levels were quantified by RT-qPCR analysis. Bars indicate means ± SD of three independent experiments, each performed in triplicate (* *p* < 0.05). (**C**) A673-1c ES cells were treated or not with doxycycline (1 μg/mL). Then, 48 h after cells were transfected with the Gli-specific construct Gli-lux, and treated or not with doxycycline for another 24 h. Bars indicate means ± SD of three independent experiments, each performed in duplicate (** *p* < 0.005). (**D**–**G**) A673-1c ES cells were treated with doxycycline (1 μg/mL) for 24 h. Ptch1 (**D**), Ptch2 (**E**), Stmn1 (**F**) and NKx2.2 (**G**) mRNA steady-state levels were quantified by RT-qPCR analysis. Bars indicate means ± SD of three independent experiments, each performed in triplicate (* *p* < 0.05; ** *p* < 0.005).

**Figure 4 cancers-12-03438-f004:**
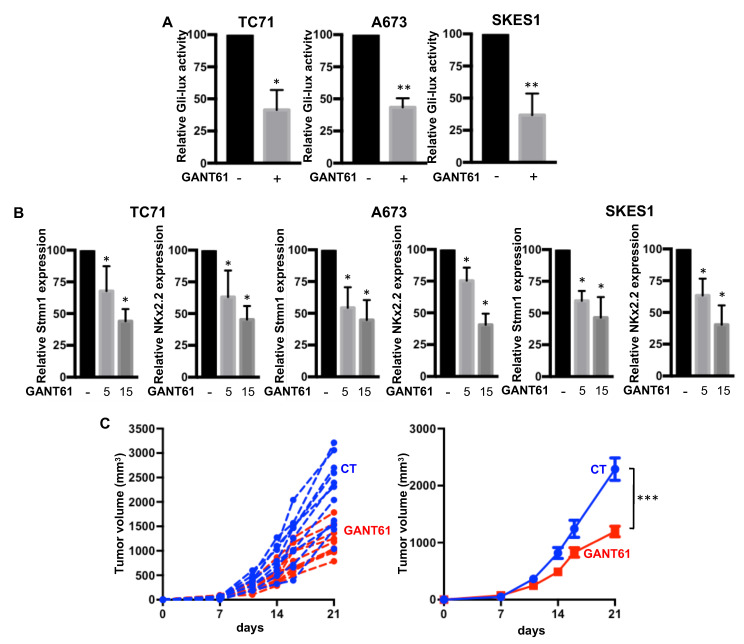
GANT61 inhibits the transcriptional activity of Gli1 and primary tumor growth in an orthotopic model of ES. (**A**) ES cells (TC71, A673 or SKES1, as indicated) were transfected with the Gli-specific construct Gli-lux. Then, 24 h after transfection, cells were treated or not with 5 µM GANT61 for 24 h. Bars indicate means ± SD of three independent experiments, each performed in duplicate (* *p* < 0.05; ** *p* < 0.005). (**B**) Stmn1 and NKx2.2 mRNA steady-state levels were quantified by RT-qPCR analysis in ES cells in the presence or absence of GANT61, as indicated (5 or 15 µM, for 24 h). Bars indicate means ± SD of three independent experiments, each performed in triplicate (* *p* < 0.05). (**C**) intramuscular paratibial injections of 1.10^6^ TC71 cells were performed in two groups of 10 nude mice treated with vehicle (blue) or GANT61 (red, 50 mg/kg), as indicated. Tumor volumes were measured two times per week for three weeks. Left panel: Individual tumor volume. Right panel: Mean ± SEM; *** *p* < 0.001.

**Figure 5 cancers-12-03438-f005:**
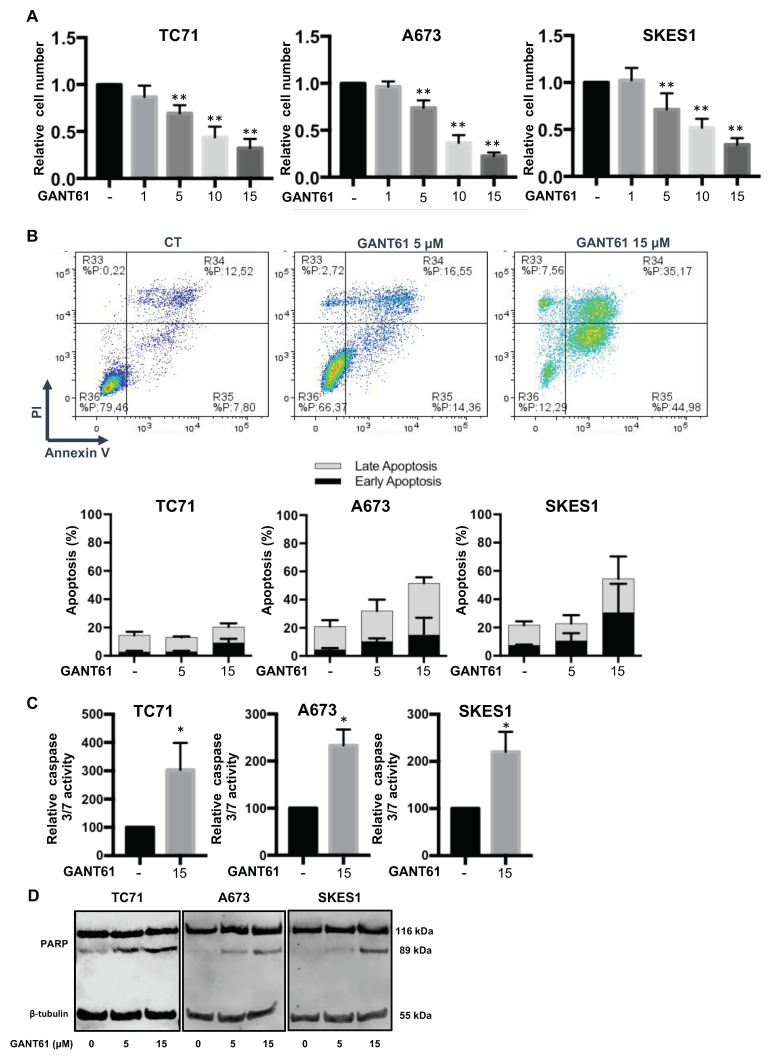
GANT61 induces in vitro cell death in ES cell lines. (**A**) three ES cell lines (TC71, A673 and SKES1) were treated or not for 48 h with GANT61, as indicated. Cell viability was evaluated as described in the material and methods section. For each cell line, the graph indicates the relative cell viability compared to untreated cells. The mean ± SD of at least five independent experiments, each performed in sextuplicate, is presented (** *p* < 0.005). (**B**) upper panels: Representative dot plots of SKES1 cells treated or not with 5 or 15 µM GANT61 for 24 h are shown (representative graphs of three experiments). Lower panels: ES cells were treated or not with 5 or 15 µM GANT61 for 24 h. Bars indicate the means ± SD of the relative number of cells in early- or late-phase apoptosis (3 independent experiments). (**C**) ES cells were treated with 15 µM GANT61 for 24 h. Relative caspase-3/7 activity was evaluated as described in the material and methods section. Bars indicate the caspase-3/7 activity (mean ± SD) of at least two independent experiments, each performed in triplicate (* *p* < 0.05). (**D**) ES cells were treated or not with 5 or 15 µM GANT61, as indicated, for 24 h. After incubation, PARP cleavage levels were detected by Western blot analysis. Representative blots of three experiments are shown.

**Table 1 cancers-12-03438-t001:** Primer sequences.

Genes	Forward	Reverse
GAPDH	TGG GTG TGA ACC ATG AGA AGT ATG	GGT GCA GGA GGC ATT GCT
B2M	TTC TGG CCT GGA GGC TAT C	TCA GGA AAT TTG ACT TTC CAT TC
GLI1	CCA ACT CCA CAG GCA TAC AGG AT	CAC AGA TTC AGG CTC ACG CTT C
GLI2	AAG TCA CTC AAG GAT TCC TGC TCA	GTT TTC CAG GAT GGA GCC ACT T
GLI3	CGC GAC TGA ACC CCA TTC TAC	GTG TTG TTG GAC TGT GTG CCA TT
Ptch1	CCC CTG TAC GAA GTG GAC ACT CTC	AAG GAA GAT CAC CAC TAC CTT GGC T
Ptch2	GAT GGG GCC ATC TCC ACA TT	CGC CGC AAA GAA GTA CCT TAC A
SMO	GCT ACT TCC TCA TCC GAG GAG TCA	GGC GCA GCA TGG TCT CGT T
STMN1	TGG TGC TCA GAG TGT GGT CA	TCA CCT GGA TAT CAG AAG AAG CCA
NKx2-2	GCA CCC CTC CTG GAG TTA GAA AC	CCA ACC CAG TGC CTC TCT CTG
EWS-FLI1	GCC AAG CTC CAA GTC AAT ATA GC	GAG GCC AGA ATT CAT GTT ATT GC

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
