# Peer review of "Sonic Hedgehog Signature in Pediatric Primary Bone Tumors: Effects of the GLI Antagonist GANT61 on Ewing’s Sarcoma Tumor Growth"

_cancers, 2020, doi:10.3390/cancers12113438_

Round 1
Reviewer 1 Report
In this study, the authors describe the efficacy of GANT61, an inhibitor of the Sonic hedgehog (SHH) signaling pathway, to counteract the development of Ewing sarcoma. First of all the authors perform an RNA seq on cell lines derived from human bone sarcoma (Ewing and Osteosarcoma) that highlighted the involvement of SHH pathway in Ewing- but not in Osteosarcoma. The efficacy of the inhibitor is demonstrated by performing the analysis of proliferation, cell cycle and caspase activity on Ewing cell lines treated with different amount of this inhibitor, and on an orthotopic mouse model. I think this is a preliminary study that has generally addressed the issues concerning the efficacy of this inhibitor. However, some points remain obscure for me. 1. The use of GANT61 has already been evaluated in Osteosarcoma, were it blocks proliferation, migration and invasion (read for example Zhang Cell Biology Int 2020) or in RMS and ES (Joon Won Yoon MBC Cancer 2020). The authors should cite and comment these studies. 2. I kindly ask the authors to explain which were the criteria for choosing TC71 to be injected into the mice, instead of the other cell lines. 3. Regarding the experiments on mice, the study describes the effect of GANT61 on the volume of the tumor mass, but there is no mention regarding the general state of health of GANT61-treated mice compared to placebo treated counterparts (weight etc etc). How did the authors chose the concentration of use of GANT61? Minor: In the Materials and Methods section, Cell Coltures, there in a typo regarding the list of the cell lines used. Some listed OS cell lines are actually the same of ES one.Author Response
We thank the reviewer for the positive feedback on our work.
Concerning the different issues.
1) Indeed, the drug GANT61 has been tested in osteosarcoma. References (Zhang and Chu, Cell Biol Int 2020 ; Joon Won Yoon MBC Cancer 2020) were added and discussed.
In the discussion section :
-) «One recent study demonstrated that the Gli1/2 inhibitor GANT61 is able to inhibit the growth of OS tumors by inducing oxidative stress via the miRNA-1286/RAB31 axis using an OS xenograft model (Zhang and Chu, Cell Biol Int. 2020).»
-) «Interestingly GANT61 seems also to be able to overcome the resistance of Rhabdomyosarcoma and ES cells to chemotherapy such as vincristine (Joon Won Yoon et al., BMC Cancer 2020).»
2) The choice of TC71 cells is purely practical, since this model is routinely used in the laboratory and mimics the human development of the disease, in particular bone degradation. The other model used in the laboratory is the injection of A673 cells, however this model presents a less important bone degradation than the first. We do not master the other models.
3) Notably, no major changes were observed following injection of the drug GANT61 (weight loss or change in mice behaviour) suggesting good tolerance to the drug. A sentence has been added in the "Materials and Methods" section. The drug concentration corresponds to the concentration used in different mice models.
4) The name of the osteosarcoma cells has been changed « HOS, KHOS, G292, MG63, U2OS, SJSA1, SAOS2 »
We hope that all the modifications made to our original manuscript address the reviewer’s issues satisfactorily and that this revised version is now acceptable for publication.
Reviewer 2 Report
This a well-designed study about the effects of the GLI antagonist GANT61 on
Osteosarcoma and Ewing’s sarcoma tumor growth. The results are presented in detail and discussed thoroughly. I believe the article meets the interest of many readers dealing with these challenging pathologies.
Author Response
We thank the reviewer for the positive feedback on our work.
Reviewer 3 Report
This is a well-written paper exploring innovative strategies regarding the treatment of osteosarcoma and Ewing’s sarcoma. The evidence provided is clear and suggest the potential role to further study Gli1 inhibition in Ewing’s sarcoma.
Few suggestions to improve the quality of this manuscript.
A general comment is to focus on authors findings which are not on HH pathway/s (as authors emphasize in in the introduction section is a complex pathway with several different mechanisms of activation and regulation), but to point out their own data that clearly demonstrate the chimeric protein role into start Gli1 overexpression and activation.
- Figure 1 shows an inconsistent activation of HH pathway in several cell lines. In particular, HH is under-expressed in 5 out of seven cell lines. Thus, rephrase manuscript focusing on authors’ data.
- Cell line EW24 seems HH-Gli1 independent (see also the bar after transfection). Therefore, it represents an ideal negative control. Do authors have data to add to confirm their findings on Gli1 inhibition in this cell line?
- Please provide in more detail some of the experiments on EW24 (e.g. Stmn1 and NKX2.2 expression). This may be added in the supplementary to gather information on Gli1 in not overexpressing tumors.
Author Response
We thank the reviewer for the positive feedback on our work.
Concerning the few suggestions to improve the quality of this manuscript.
1) As suggested, we have focused now the discussion on the major role of the fusion protein EWS-Fli1 in the activation of the SHH pathway in Ewing's sarcoma..
In the discussion section : « Our work reinforces the crucial role of EWS-Fli1 in the development of ES as previously demonstrated. Indeed, the reduction of EWS-FLi1 expression by small interfering RNA significantly reduces the proliferative, invasive and tumorigenic phenotype of ES, both in vitro and in vivo (Tanaka et al., J Clin Invest.1997 ; Pishas and Lessnick, F1000 Research, 2016). Thus, since the transcriptional activity of Gli1 is directly activated by EWS-Fli1 in ES, it appears crucial to directly target the transcription factor Gli1, rather than targeting the SMO receptor, using cyclopamine, for example.».
2) For Figure 1, changes have been made to the text as requested. In the results section « Note that the EW24 ES cells appear to have lower Gli1 expression than the other ES cell lines (Figure 1A). However, the data obtained using RT-qPCR indicate that the Gli1 expression in this cell line remains higher than those measured in all the OS lines except the U2OS cells.»
3) Regarding EW24 cell lines : The RT-qPCR analysis clearly demonstrates that even for EW24 ES cells that express Gli1 more weakly, the level of Stmn1 and NKx2.2 expression remains higher than that measured in OS cells (Figure 2C). A sentence has been added in the text.
We hope that all the modifications made to our original manuscript address the reviewer’s issues satisfactorily and that this revised version is now acceptable for publication